# A Bayesian Motivated Two-Sample Test Based on Kernel Density Estimates

**DOI:** 10.3390/e24081071

**Published:** 2022-08-03

**Authors:** Naveed Merchant, Jeffrey D. Hart

**Affiliations:** Department of Statistics, Texas A&M University, College Station, TX 77840, USA; hart@stat.tamu.edu

**Keywords:** Bayes factors, permutation tests, cross-validation, consistent tests, Kolmogorov–Smirnov test

## Abstract

A new nonparametric test of equality of two densities is investigated. The test statistic is an average of log-Bayes factors, each of which is constructed from a kernel density estimate. Prior densities for the bandwidths of the kernel estimates are required, and it is shown how to choose priors so that the log-Bayes factors can be calculated exactly. Critical values of the test statistic are determined by a permutation distribution, conditional on the data. An attractive property of the methodology is that a critical value of 0 leads to a test for which both type I and II error probabilities tend to 0 as sample sizes tend to *∞*. Existing results on Kullback–Leibler loss of kernel estimates are crucial to obtaining these asymptotic results, and also imply that the proposed test works best with heavy-tailed kernels. Finite sample characteristics of the test are studied via simulation, and extensions to multivariate data are straightforward, as illustrated by an application to bivariate connectionist data.

## 1. Introduction

Ref. [1] proposed the use of cross-validation Bayes factors in the classic two-sample problem of comparing two distributions. Their basic idea is to randomly divide the data into two distinct parts, call them *A* and *B*, and to define two models based on kernel density estimates from part *A*. One model assumes that the two distributions are the same and the other allows them to be different. A Bayes factor comparing the two part *A* models is then defined from the part *B* data. In order to stabilize the Bayes factor, Ref. [1] suggest that a number of different random data splits be used, and the resulting log-Bayes factors averaged.

In the current paper we consider a special case of this approach in which the part *A* data consists of all the available observations save one. If the sample sizes of the two data sets are *m* and *n*, this entails that a total of m+n log-Bayes factors may be calculated. The average of these m+n quantities becomes the test statistic here considered, and is termed ALB.

Although ALB is an average of log-Bayes factors, it does not lead to a consistent Bayes test because each of the log-Bayes factors is based on just a single observation. Ref. [1] suppose that the validation set size grows to *∞*, while in our case it remains of size 1. This results in the ALB converging to the Kullback–Leibler divergence of the two densities, and not *∞* as in the case of [1]. We therefore use frequentist ideas to construct our test. The exact null distribution of ALB conditional on order statistics is obtained using permutations of the data. Doing so leads to a consistent frequentist test whose size is controlled exactly. The problem of bandwidth selection is dealt with by using leave-one-out likelihood cross-validation applied to the combination of the two data sets. This method is computationally efficient in that the resulting bandwidth is invariant to permutations of the combined data, and therefore has to be computed just once. Our methodology is easily extended to bivariate data, and we do so in a real data example.

Ref. [2] also use a permutation test based on kernel estimates for the two-sample problem, their statistic being based on an $L_{2}$ distance. Ref. [3] shows how other distances and divergences compare when applying them to the general *k*-sample problem, restricting their comparisons to the one-dimensional case. Our method mainly differs from these procedures by virtue of its Bayesian motivation. Existing methodology that most closely resembles ours is that of [4], who use a kernel-based marginal likelihood ratio to test goodness of fit of parametric models for a distribution. Their marginal likelihood employs a prior for a bandwidth, as does ours.

## 2. Methodology

We assume that $X=(X_{1},\ldots ,X_{m})$ are independent and identically distributed (i.i.d.) from density *f*, and independently $Y=(Y_{1},\ldots ,Y_{n})$ are i.i.d. from density *g*. We are interested in the problem of testing the null hypothesis that *f* and *g* are identical on the basis of the data X and Y. Let $U=(U_{1},\ldots ,U_{k})$ be an arbitrary set of *k* scalar observations, and define a kernel density estimate by
$${\hat{f}}_{K}\left(u|h,U\right)=\frac{1}{kh}\sum_{i=1}^{k}K\left(\frac{u-U_{i}}{h}\right),\;-\infty <u<\infty ,$$
where *K* is the kernel and h>0 the bandwidth.

### 2.1. The Test Statistic

Let $Z_{i}=X_{i}$, i=1,…,m, $Z_{i}=Y_{i-m}$, i=m+1,…,m+n, $Z=(Z_{1},\ldots ,Z_{m+n})$ and $Z_{i}$ be the vector Z with all its components except $Z_{i}$, i=1,…,m+n. Furthermore, let $X_{i}$ be all the components of X except $X_{i}$, i=1,…,m, and $Y_{j}$ all the components of Y except $Y_{j}$, j=1,…,n. If we assume that *f* is identical to *g*, then potential models for *f* are $M_{0i}=\left\{{\hat{f}}_{K}\left(\;\cdot \;|h,Z_{i}\right):h>0\right\}$, i=1,…,m+n. Suppose that 1≤i≤m. If we allow that *f* and *g* are different, then a model for the datum $Z_{i}$ is $M_{1i}=\left\{{\hat{f}}_{K}\left(\;\cdot \;|a,X_{i}\right):a>0\right\}$. In this case a legitimate Bayes factor for comparing $M_{0i}$ and $M_{1i}$ on the basis of the datum $Z_{i}$ has the form
$$B_{i}=\frac{\int_{0}^{\infty }\pi \left(a\right){\hat{f}}_{K}\left(Z_{i}|a,X_{i}\right)\;da}{\int_{0}^{\infty }\pi \left(h\right){\hat{f}}_{K}\left(Z_{i}|h,Z_{i}\right)dh},\;i=1,\ldots ,m,$$
where, mainly for convenience, we have assumed that the bandwidth priors are the same in all cases. Likewise, if i=m+1,…,m+n, then $M_{1i}=\left\{{\hat{f}}_{K}\left(\;\cdot \;|b,Y_{i-m}\right):b>0\right\}$ is a model for the datum $Z_{i}$, and a Bayes factor for comparing $M_{0i}$ and $M_{1i}$ is
$$B_{i}=\frac{\int_{0}^{\infty }\pi \left(a\right){\hat{f}}_{K}\left(Z_{i}|a,Y_{i-m}\right)\;da}{\int_{0}^{\infty }\pi \left(h\right){\hat{f}}_{K}\left(Z_{i}|h,Z_{i}\right)dh},\;i=m+1,\ldots ,m+n.$$

When *m* and *n* are large, it is expected that $M_{1i}$ will be a good model for *f* if i=1,…,m and for *g* if i=m+1,…,m+n. Likewise, each of $M_{0i}$ will be a good model for the common density on the assumption that *f* and *g* are identical. However, none of $B_{1},\ldots ,B_{m+n}$ will be Bayes factors that can provide convincing evidence for either hypothesis simply because each one uses likelihoods based on a single datum. At first blush one might think that a solution to this problem is to take the average of the m+n log-Bayes factors:(1)$$\begin{array}{c} ALB=\frac{1}{\left(m+n\right)}\sum_{i=1}^{m+n}\log B_{i}. \end{array}$$However, this results in a statistic that will consistently estimate 0 or a positive constant in the respective cases f≡g or f≢g. In neither case does the statistic have the property of Bayes consistency, i.e., the property that the Bayes factor tends to 0 and *∞* when f≡g and f≢g, respectively.

The discussion immediately above points out a fundamental fact that seems not to have been widely discussed: combining a large number of inconsistent Bayes factors does not necessarily lead to a consistent Bayes factor. A guiding principle in [1] was that of averaging log-Bayes factors from different random splits of the data with the aim of producing a more stable log-Bayes factor. However, in order for this practice to yield a *consistent* Bayes factor, it is important that each of the log-Bayes factors being averaged is consistent. Furthermore, to ensure this consistency, it is necessary that the sizes of both the training and validation sets tend to *∞* with the samples sizes *m* and *n*. Obviously this is not the case when the size of each validation set is just 1, as in the current paper.

An advantage of the approach proposed herein is that the practitioner does not have to choose the size of the training sets. The cost is that the resulting statistic does not have the property of Bayes consistency. We thus propose that the statistic be used in frequentist fashion. An appealing way of doing so is to use a permutation test, which (save for certain practical issues to be discussed) leads to a test with exact type I error probability for all m>1 and n>1. Let $Z_{\left(1\right)}<Z_{\left(2\right)}<\cdots <Z_{\left(m+n\right)}$ be the order statistics for the combined sample. Let $j=(j_{1},\ldots ,j_{m+n})$ be a random permutation of 1,…,m+n, and define T(j) to be the statistic (Equation 1) when the *X*-sample is taken to be $Z_{j_{1}},\ldots ,Z_{j_{m}}$ and the *Y*-sample to be $Z_{j_{m+1}},\ldots ,Z_{j_{m+n}}$. It follows that, conditional on the order statistics $Z_{\left(1\right)},\ldots ,Z_{\left(m+n\right)}$, the (m+n)! values taken on by T(·) are equally likely. Therefore, if $t_{m,n}$ is a 1−α quantile of the empirical distribution of T(·), then the test that rejects f≡g when $T\geq t_{m,n}$ will have an (unconditional) type I error probability of α. As will be shown in the Section A.3, ALB is negative with probability tending to 1 as m,n→∞, implying that for any α>0 $t_{m,n}$ will be negative for *m* and *n* large enough. From an evidentiary standpoint, it is nonsense to reject $H_{0}$ for a negative value of ALB. We therefore suggest using the critical value $\max(0,t_{m,n})$, which ensures that the test is sensible and has level α.

### 2.2. The Effect of Using Scale Family Priors

Let $\pi_{0}$ be an arbitrary density with support (0,∞). A possible family of priors is one that contains all rescaled versions of $\pi_{0}$. For b>0, using the prior $\pi \left(h\right)=\pi_{0}\left(h/b\right)/b$ and making the change of variable h/b=u in the denominator of $B_{i}$, we have
$$\begin{array}{c} \int_{0}^{\infty }b^{-1}\pi_{0}\left(h/b\right){\hat{f}}_{K}\left(Z_{i}|h,Z_{i}\right)dh={\hat{f}}_{L}\left(Z_{i}|b,Z_{i}\right), \end{array}$$
where the kernel *L* is
(2)$$\begin{array}{c} L\left(z\right)=\int_{0}^{\infty }u^{-1}\pi_{0}\left(u\right)K\left(z/u\right)\;du,\;\mathrm{for}\;\mathrm{all}\;z. \end{array}$$So, by using this type of prior, each marginal likelihood comprising ALB becomes a kernel density estimate with bandwidth equal to the scale parameter of the prior. In one sense this is disappointing since it means that averaging kernel estimates with respect to a bandwidth prior does not actually sidestep the issue of choosing a smoothing parameter. One has simply traded bandwidth choice for choice of the prior’s scale. However, it turns out that there is a quantifiable advantage to using a prior for the bandwidth of *K*. As detailed in the Section A.2, likelihood cross-validation is often more efficient when applied to ${\hat{f}}_{L}$ rather than to ${\hat{f}}_{K}$.

When using a scale family of priors, the result immediately above implies that
(3)$$\begin{array}{ccc} (m+n)ALB & = & \sum_{i=1}^{m}\log\left({\hat{f}}_{L}\left(X_{i}|b,X^{i}\right)\right)+\sum_{j=1}^{n}\log\left({\hat{f}}_{L}\left(Y_{j}|b,Y^{j}\right)\right)\\ & & -\sum_{i=1}^{m+n}\log\left({\hat{f}}_{L}\left(Z_{i}|b,Z^{i}\right)\right), \end{array}$$
and so the proposed statistic is proportional to the log of a likelihood ratio. The two likelihoods are cross-validation likelihoods, and the numerator and denominator of the ratio correspond to the hypotheses of different and equal densities, respectively.

In practice one must select both the kernel *L* and bandwidth *b*. For the moment we assume that *L* is given. The denominator of exp((m+n)ALB) as a function of *b* is the likelihood cross-validation criterion, as studied by [5], based on the combined sample. We propose using $b=\hat{b}$, the maximizer of this denominator. This bandwidth has the desirable property that it is invariant to the ordering of the data in the combined sample. Let $ALB^{*}$ be the value of test statistic (Equation 1) for a permuted data set. One should use the principle that $ALB^{*}$ is the same function of the permuted data as ALB is of the original data. So, in principle the bandwidth should be selected for every permuted data set, but because of the invariance of $\hat{b}$ to the ordering of the combined sample, this data-driven bandwidth equals $\hat{b}$ for every permuted data set. This results in a large computational savings relative to a procedure that selects the bandwidth differently for the *X*- and *Y*-samples. Using the same bandwidth under both null and alternative hypotheses also fits with the principle espoused by [6].

Concerning *L*, Ref. [5] showed that kernels must be relatively heavy-tailed in order for them to perform well with respect to likelihood cross-validation. In particular, he shows that likelihood cross-validation fails miserably as a method for choosing the bandwidth of a kde based on a Gaussian kernel. The tails of the kernel must be considerably heavier than those of a Gaussian density in order for likelihood cross-validation to be effective. Proposition A1 in the Section A.1 shows that under very general conditions *L* (as defined in (Equation 2)) has heavier tails than those of *K*. Therefore, the Bayesian notion of averaging commonly used kernel estimates with respect to a prior brings the resulting kernel estimate more in line with the conditions of [5]. This has a substantial benefit for our statistic inasmuch as we use a likelihood cross-validation bandwidth in its construction.

Consider the following kernel proposed by [5]:$$L_{0}\left(u\right)=\frac{1}{\sqrt{8\pi e}\;\Phi \left(1\right)}\exp\left[-\frac{1}{2}{\left(\log(1+|u|)\right)}^{2}\right].$$Suppose that a kde is defined using kernel $L_{0}$ and its bandwidth is chosen by likelihood cross-validation. Ref. [5] shows that, in general, this cross-validation bandwidth will be asymptotically optimal in a Kullback–Leibler sense. We will therefore use $L_{0}$ in all subsequent simulations. Results in the Section A.2 provide a kernel *K* and corresponding prior that produce $L_{0}$.

### 2.3. Further Properties of ALB

In the Section A.3 we will show that the ALB test is consistent in the frequentist sense. In other words, for any alternative the power of an ALB test of fixed level tends to 1 as *m* and *n* tend to *∞*.

Interestingly, ALB has the property of being sharply bounded above. It can be rewritten as follows: $$\sum_{i=1}^{m}\log\left({\hat{f}}_{L}\left(X_{i}|b,X^{i}\right)/{\hat{f}}_{L}\left(X_{i}|b,Z^{i}\right)\right)+\sum_{j=1}^{n}\log\left({\hat{f}}_{L}\left(Y_{j}|b,Y^{j}\right)/{\hat{f}}_{L}\left(Y_{j}|b,Z^{m+j}\right)\right).$$Defining $p_{m,n}=\left(m-1\right)/\left(m+n-1\right)$,
$${\hat{f}}_{L}\left(X_{i}|b,Z^{i}\right)=p_{m,n}{\hat{f}}_{L}\left(X_{i}|b,X^{i}\right)+\left(1-p_{m,n}\right){\hat{f}}_{L}\left(X_{i}|b,Y\right),\;i=1,\ldots ,m,$$
and therefore
$$\begin{array}{c} \frac{{\hat{f}}_{L}\left(X_{i}|b,X^{i}\right)}{{\hat{f}}_{L}\left(X_{i}|b,Z^{i}\right)}=\frac{1}{p_{m,n}}\cdot \frac{p_{m,n}{\hat{f}}_{L}\left(X_{i}|b,X^{i}\right)}{p_{m,n}{\hat{f}}_{L}\left(X_{i}|b,X^{i}\right)+\left(1-p_{m,n}\right){\hat{f}}_{L}\left(X_{i}|b,Y\right)}\leq \frac{1}{p_{m,n}}. \end{array}$$A similar bound applies for the other component of ALB, implying that
(4)$$\begin{array}{c} ALB\leq -\left[\left(\frac{m}{m+n}\right)\log\left(p_{m,n}\right)+\left(1-\frac{m}{m+n}\right)\log\left(\frac{n-1}{m+n-1}\right)\right]. \end{array}$$Using the fact that −[xlog(x)+(1−x)log(1−x)] has its maximum at x=1/2 when 0≤x≤1, bound (Equation 4) implies that
$$ALB\leq \log\left(2\right)\cdot \max\left(\frac{m}{\left(m-1\right)},\frac{n}{\left(n-1\right)}\right).$$Unless one of *m* and *n* is very small, the effective bound on ALB is log(2). This reinforces the fact that ALB does not have the property of Bayes consistency. While it is true that ALB is an average of Bayes factors, none of these Bayes factors can ever provide compelling evidence in favor of the alternative. To reiterate, this problem is overcome by employing ALB in frequentist fashion.

While ALB can take on positive values when the null hypothesis is true, our proof of frequentist consistency shows that, under $H_{0}$, P(ALB<0)→1 as m,n→∞. This implies that if 0 is used as a critical value, then the resulting test level tends to 0 as m,n→∞. So, even though |ALB| does not tend to *∞*, the *sign* of ALB provides compelling evidence for the hypotheses of interest when the sample sizes are large.

The exact conditional distribution of ALB is known under the null hypothesis, as we use a permutation test. Nonetheless, it is of some interest to have an impression of the *unconditional* distribution of ALB. To this end, we randomly select two normal mixture densities that differ. The number of components *M* in the first mixture is between 2 and 20 and chosen from a distribution such that the probability of *m* is proportional to $m^{-1}$, m=2,…,20. Given M=m, mixture weights are drawn from a Dirichlet distribution with all *m* parameters equal to 1/2. Given M=m and mixture weights, variances $\sigma_{1}^{2},\ldots ,\sigma_{m}^{2}$ of the normal components are a random sample from an inverse gamma distribution with both parameters equal to 1/2. Finally, means $\mu_{1},\ldots ,\mu_{m}$ of the normal components are such that $\mu_{1},\ldots ,\mu_{m}$ given $\sigma_{1},\ldots ,\sigma_{m}$ are independent with $\mu_{j}|\sigma_{j}∼N\left(0,\sigma_{j}^{2}\right)$, j=1,…,m. The second normal mixture is independently selected using exactly the same mechanism. Random selection of densities in this manner for simulation studies has been proposed and explored in [7].

We draw a sample of size 100 from each of the two randomly generated densities (so that m=n=100), and then compute ALB. This procedure is replicated on the same two densities 100 times. After this, we repeat the whole procedure for nine more pairs of randomly selected densities. The results are seen in Figure 1. Save for case 3, the proportion of positive ALBs is nearly 1 in all cases.

We repeated a similar procedure for the null hypothesis setting. The simulation was exactly the same except that in each of the ten cases, only one density was generated, and a pair of independent samples (of size 100 each) was selected from this same density. The resulting ALB distributions can be seen in Figure 2. The proportion of the cases where ALB<0 for the 10 densities were, respectively, 0.89, 0.83, 0.83, 0.84, 0.85, 0.87, 0.91, 0.84, 0.84, and 0.76. These results are consistent with the fact that P(ALB<0) tends to 1 with sample size.

We feel that ALB has potential for screening variables in a binary classification problem. Since ALB is negative with high probability under $H_{0}$, we feel that 0 is a nicely interpretable cutoff for variable inclusion. However, we leave this topic for future research.

## 3. Simulations

We perform a small simulation study to investigate the size and power of our test. To explore the effect of the number of permutations, we generate 500 pairs of data sets, with one data set being a random sample of size m=50 from a standard normal distribution, and the other a random sample of size n=50 from a normal distribution with mean 0 and standard deviation 2. For each of the 500 pairs of data sets, the 95th percentile of ALBs is approximated using a range of different numbers (*N*) of permutations starting at 100 and increasing by a factor of 1.5 up to 3845. Results are indicated by the boxplots in Figure 3. The percentiles are centered at approximately the same value for all *N*. Not surprisingly, the variability of the percentiles becomes smaller as *N* increases. This implies a certain amount of mismatch between percentiles at N=3845 and those at smaller *N*.

The consequence of the mismatch just alluded to can be investigated by determining the true conditional and unconditional levels of tests based on small *N*. For the null case, two data sets, each of size 50, are generated from a common normal distribution. Since the distribution of ALB is invariant to location and scale in the null case, we use a standard normal without loss of generality. For each pair of data sets, the data are randomly permuted 338 times, which leads to 338 values of ALB. A second set of 3845 permutations is then performed, leading to 3845 more values of ALB. The proportion of ALBs from the second set that exceed the 95th percentile of the ALBs formed from the first set is then determined. This proportion is approximately equal to the conditional level of the test based on 338 permutations. This same procedure is used for each of 500 data sets, and the resulting distribution of approximate levels is shown in Figure 4.

The histogram is centered near 0.05, and 87% of the conditional levels are between 0.03 and 0.07. Furthermore, an approximation to the unconditional level is $\sum_{i=1}^{500}{\hat{\alpha }}_{i}/500=0.053$, where ${\hat{\alpha }}_{i}$ is the approximate conditional level for the *i*th data set, i=1,…,500. Based on these results, use of only 338 permutations is arguably adequate.

The same experiment is repeated except now the two data sets are drawn from different distributions, a standard normal and a normal with mean 0 and standard deviation 2. Results from this experiment are given in Figure 5.

As in the null case, the conditional levels based on the use of 338 permutations are quite good. Eighty-eight percent of the levels are between 0.03 and 0.07, and the approximate unconditional level is 0.051.

The proportion of ALBs from permuted data sets that are larger than the ALB computed from the original data provides a *p*-value. The *p*-values obtained with our method (based on 3845 permutations) are compared to the *p*-values obtained with the Kolmogorov–Smirnov test and Bowman’s two-sample test. Results are summarized in Figure 6 and Figure 7. In 98% of the replications the K-S *p*-value was larger than the ALB *p*-value, and in 57% of the cases the Bowman *p*-value was equal to or larger than the ALB *p*-value. These results suggest that in this case our test has much better power than that of the Kolmogorov–Smirnov test and power at least comparable to that of Bowman’s test.

## 4. A Bivariate Extension of the Two-Sample Test and Application to Connectionist Bench Data

Our method can be extended to the bivariate case by using a bivariate kernel density estimate. Assume now that $X=(X_{1},...,X_{m})$ are independent and identically distributed from density *f* and $Y=(Y_{1},...,Y_{m})$ are independent and identically distributed from *g*, where $X_{i}$ and $Y_{j}$ are each bivariate observations, i=1,…,m, j=1,…,n.

A product kernel *K* will be used, i.e., the bivariate kernel *K* is the product of two univariate kernels. For *k* arbitrary bivariate observations $U=(U_{1},\ldots ,U_{k})$, $U_{i}=\left(U_{i1},U_{i2}\right)$, i=1,…,k, and $u=(u_{1},u_{2})$, the kernel estimate is defined by
$${\hat{f}}_{K}\left(u|h,U\right)=\frac{1}{kh_{1}h_{2}}\sum_{i=1}^{k}K\left(\frac{u_{1}-U_{i1}}{h_{1}}\right)K\left(\frac{u_{2}-U_{i2}}{h_{2}}\right),$$
where $-\infty <u_{1}<\infty$, $-\infty <u_{2}<\infty$ and $h=(h_{1},h_{2})$ is a two-vector of (positive) bandwidths.

We will use the same sort of notation as before, i.e., $Z_{i}=X_{i}$, i=1,…,m, $Z_{i}=Y_{i-m}$, i=m+1,…,m+n, $Z=(Z_{1},\ldots ,Z_{m+n})$ and $Z_{i}$ is the object Z with all its components except $Z_{i}$, i=1,…,m+n. In this case the *i*th Bayes factor is defined as
$$B_{i}=\frac{\int_{0}^{\infty }\int_{0}^{\infty }\pi \left(h_{1},h_{2}\right){\hat{f}}_{K}\left(Z_{i}|h,X_{i}\right)\;dh_{1}dh_{2}}{\int_{0}^{\infty }\int_{0}^{\infty }\pi \left(h_{1},h_{2}\right){\hat{f}}_{K}\left(Z_{i}|h,Z_{i}\right)dh_{1}dh_{2}},\;i=1,\ldots ,m,$$
and similarly for i=m+1,…,m+n. As before the test statistic is $ALB=\sum_{i=1}^{m+n}\log B_{i}/\left(m+n\right)$.

This form may seem daunting, but reduces to a more familiar form if we take $\pi \left(h_{1},h_{2}\right)=\pi_{0}\left(h_{1}/b_{1}\right)\pi_{0}\left(h_{2}/b_{2}\right)/\left(b_{1}b_{2}\right)$. In this case, proceeding exactly as in Section 2, $B_{i}$ has the form
$$B_{i}=\frac{{\hat{f}}_{L}\left(Z_{i}|b,X_{i}\right)}{{\hat{f}}_{L}\left(Z_{i}|b,Z_{i}\right)},\;i=1,\ldots ,m,$$
and similarly for i=m+1,…,m+n, where $b=(b_{1},b_{2})$ and *L* is defined by (Equation 2).

We will analyze a subset of the connectionist bench data, which consist of measurements obtained after bouncing sonar waves off of either rocks or metal cylinders. The data may be found at the UCI Machine Learning repository, Ref. [8]. There are 60 variables in the data set, with m=111 and n=97 measurements of each variable for the metal cylinders and rocks, respectively. Variable numbers (1 to 60) correspond to increasing aspect angles at which signals are bounced off of either metal or rock, and each of the 60 numbers is an amount of energy within a particular frequency band, integrated over a certain period of time. We will apply our test to see if the first two variables (corresponding to the smallest aspect angles) have a different distribution for rocks than they do for metal cylinders. In our analysis *K* is taken to be ϕ, the standard normal density, and $\pi_{0}$ to be of the form (Equation 5). In this event *L* is a *t*-density with ν degrees of freedom. We will use ν=3, leading to a fairly heavy-tailed kernel, which is desirable for reasons discussed previously.

The data for each variable are inherently between 0 and 1, and bivariate kernel estimates display boundary effects along the lines x=0 and y=0, with the largest bias near the origin. We therefore use a reflection technique to reduce bias along these two lines. Suppose one has *k* observations $\left(x_{1},y_{1}\right),\ldots ,\left(x_{k},y_{k}\right)$ on the unit square. Each observation $\left(x_{i},y_{i}\right)$ is reflected to create three new observations: $\left(x_{i},-y_{i}\right)$, $\left(-x_{i},-y_{i}\right)$ and $\left(-x_{i},y_{i}\right)$, i=1,…,k. One then simply computes, at points in the unit square, a standard kernel density estimate from the data set of size 4k, and multiplies it by 4 to ensure integration to 1. The value of ALB is computed as described previously except that each leave-out estimate leaves out four values: the observation at which the estimate is evaluated plus its three reflected versions. In this way the kde is constructed from data that are independent of the value at which the kde is evaluated.

Kernel density estimates for variables 1 and 2 in the form of heat maps are shown in Figure 8 and Figure 9, and contours of the estimates are given in Figure 10. The latter figure suggests that the distributions for metal cylinders and rock are different. The value of ALB turned out to be 0.013, and an approximate *p*-value based on 10,000 permuted data sets was 0.0076. So, there is strong evidence of a difference between the rock and metal bivariate distributions. Interestingly, the percentage of negative ALBs among the 10,000 permutations was 0.9785. A kernel density estimate based on the 10,000 values of $ALB^{*}$ is shown in Figure 11.

## 5. Conclusions and Future Work

We have proposed a new nonparametric test of the null hypothesis that two densities are equal. An attractive property of the test is that its critical values are defined by a permutation distribution, allaying essentially any concern about test validity. The fact that the statistic is an average of log-Bayes factors leads to another attractive property: a critical value of 0 leads to a test with type I error probability tending to 0 with sample size. A simulation study showed the new test to have much better power than the Kolmogorov–Smirnov test in a case where the two densities differed with respect to scale. An application to connectionist data illustrated the usefulness of our methodology for bivariate data.

Future work includes efforts to increase the speed of computing the test statistic and its permutation distribution, especially for large data sets. We are also interested in applying the new test to the problem of screening variables prior to performing binary classification. A common method of doing so is to compute a two-sample test statistic for each variable, and to then select variables whose statistics exceed some threshold. An inherent problem in this approach is objectively choosing a threshold. Results of the current paper suggest that 0 would be a natural and effective threshold for variable screening.

## Figures and Tables

**Figure 1 entropy-24-01071-f001:**
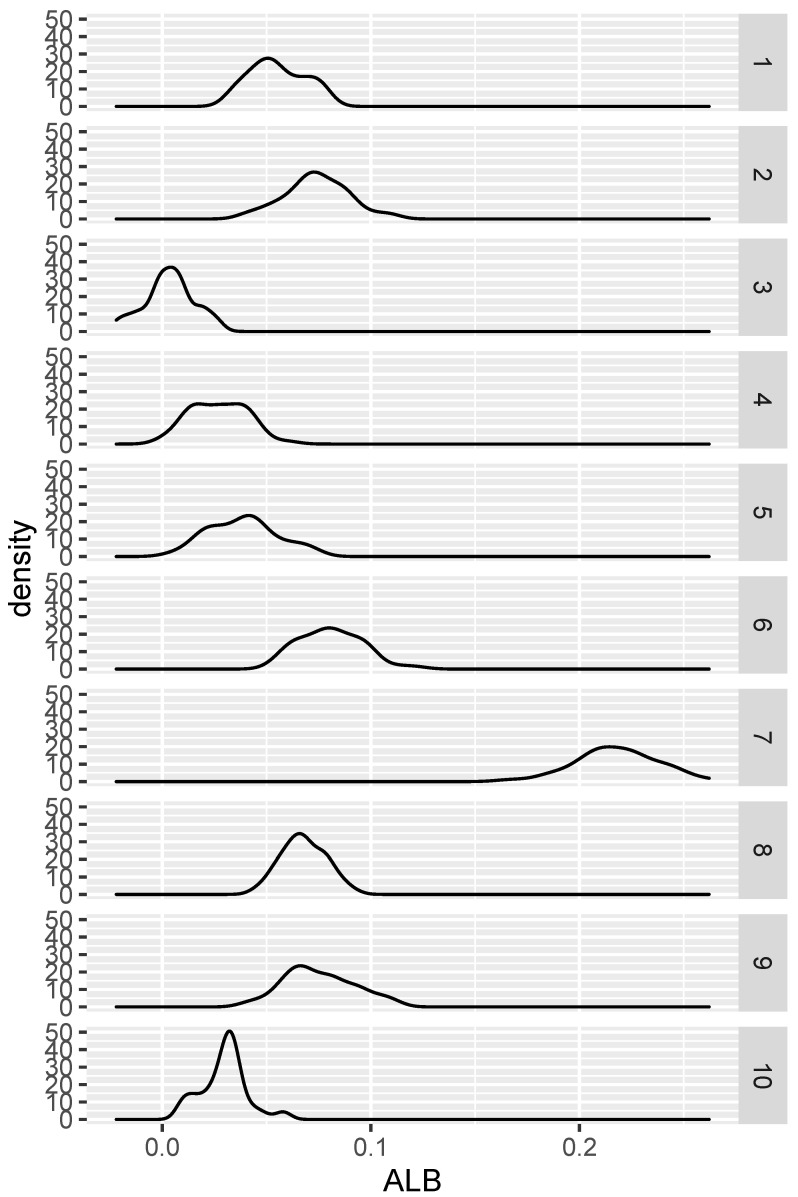
Distribution of ALB under various alternative hypotheses.

**Figure 2 entropy-24-01071-f002:**
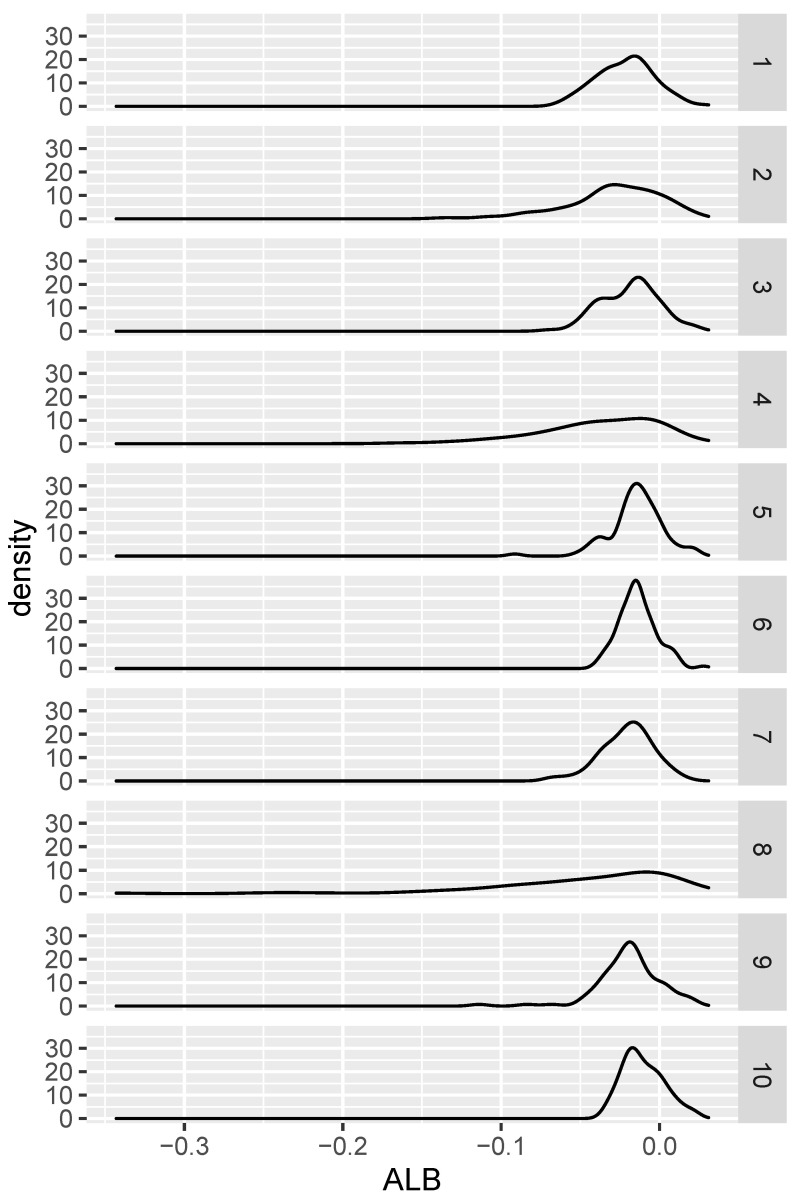
Distribution of ALB under various null hypotheses.

**Figure 3 entropy-24-01071-f003:**
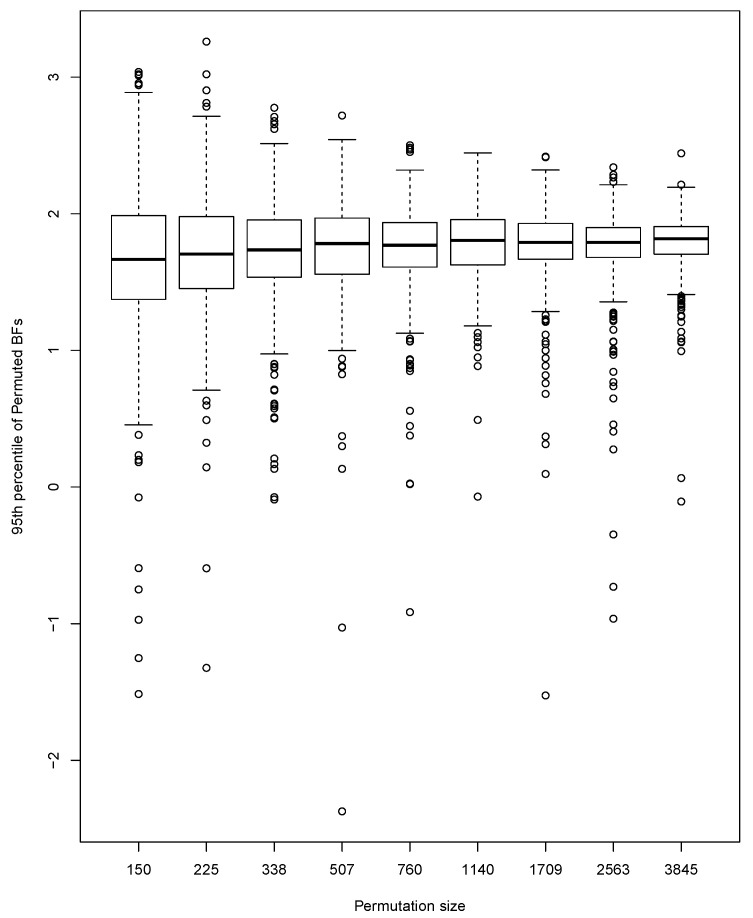
Effect of number of permutations on the 95th percentile of permutation distributions.

**Figure 4 entropy-24-01071-f004:**
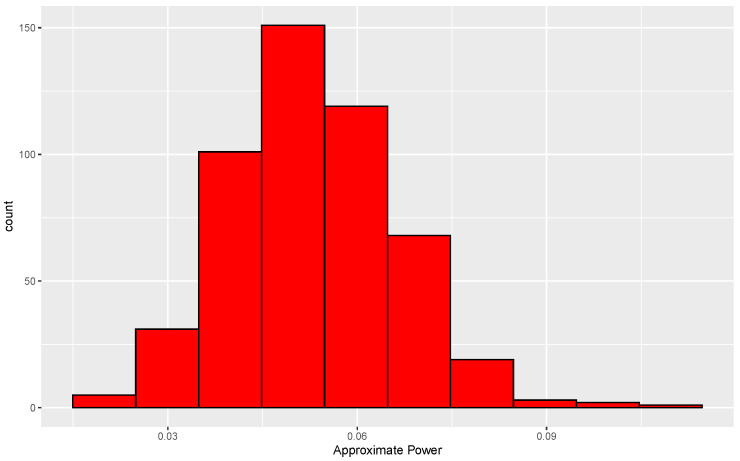
Distribution of approximate conditional levels of permutation tests under the null hypothesis. Each conditional level is the proportion of 3845 ALBs from permuted data sets that exceed the 95th percentile of ALBs formed from 338 permuted data sets. Results are based on 500 replications in each of which both distributions are standard normal.

**Figure 5 entropy-24-01071-f005:**
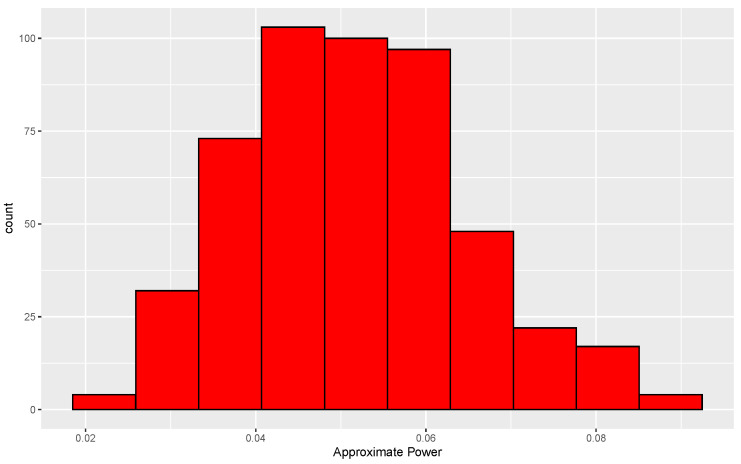
Distribution of approximate conditional levels of permutation tests under an alternative hypothesis. Each conditional level is the proportion of 3845 ALBs from permuted data sets that exceed the 95th percentile of ALBs formed from 338 permuted data sets. Results are based on 500 replications in each of which one distribution is standard normal and the other is normal with mean 0 and standard deviation 2.

**Figure 6 entropy-24-01071-f006:**
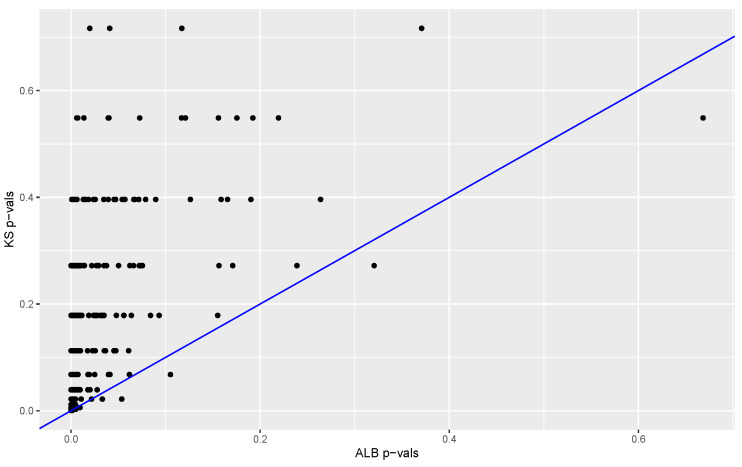
Kolmogorov–Smirnov *p*-values versus ALB *p*-values. Results are based on 500 data sets in each of which one distribution is standard normal and the other is normal with mean 0 and standard deviation 2. The ALB *p*-value is less than the KS-test *p*-value in 98% of cases. There are only 183 *p*-values from the KS-test that are less than 0.05.

**Figure 7 entropy-24-01071-f007:**
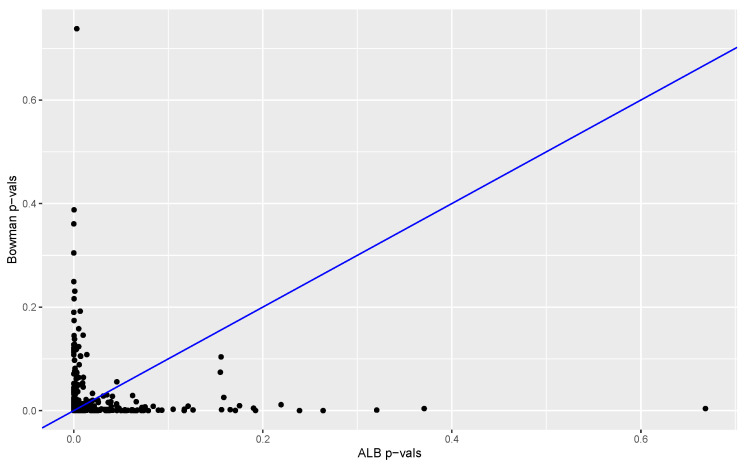
Bowman *p*-values versus ALB *p*-values. Results are based on 500 data sets in each of which one distribution is standard normal and the other is normal with mean 0 and standard deviation 2. The number of *p*-values less than 0.05 for Bowman’s test and the ALB test are 454 and 458, respectively. The ALB *p*-value is less than, more than and equal to the Bowman *p*-value in 49%, 43% and 8% of cases, respectively.

**Figure 8 entropy-24-01071-f008:**
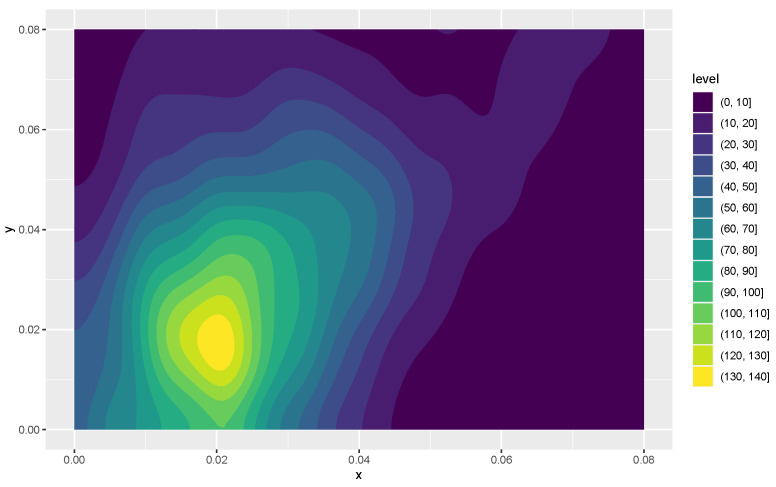
A heat map of the first two variables for the signals bounced off the metal cylinder. Variables *x* and *y* correspond to the smallest and next to smallest aspect angles, respectively.

**Figure 9 entropy-24-01071-f009:**
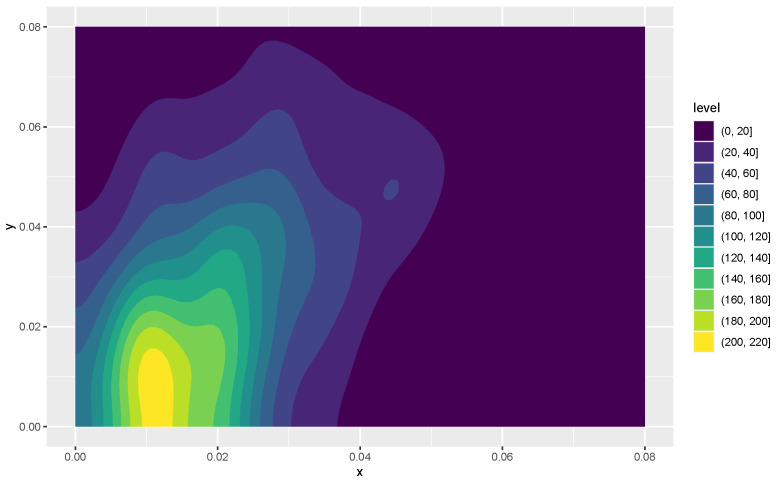
A heat map of the first two variables for the signals bounced off the rock object. Variables *x* and *y* are as defined in Figure 8.

**Figure 10 entropy-24-01071-f010:**
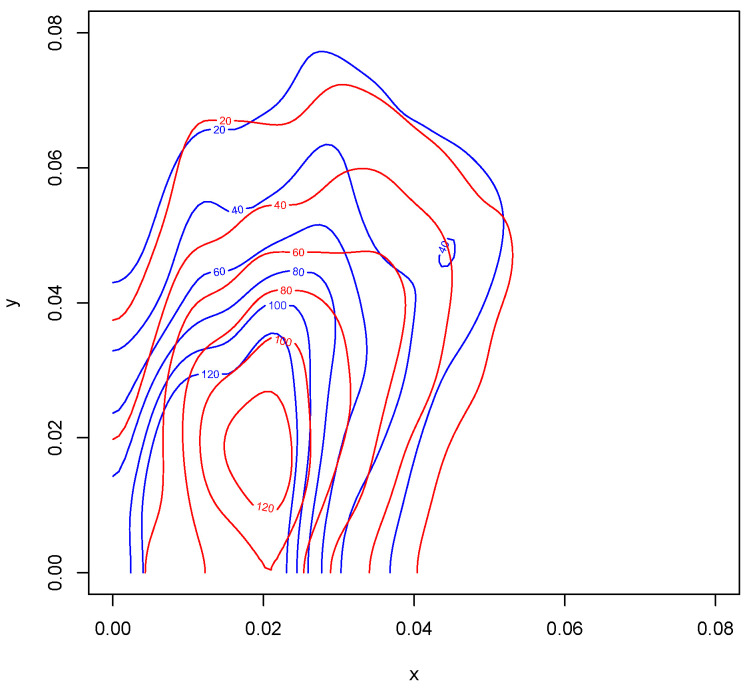
Contour plots of the first two variables of both rock and cylinder objects. The blue contours correspond to the rock measurements and red to the cylinder measurements. Variables *x* and *y* are as defined in Figure 8.

**Figure 11 entropy-24-01071-f011:**
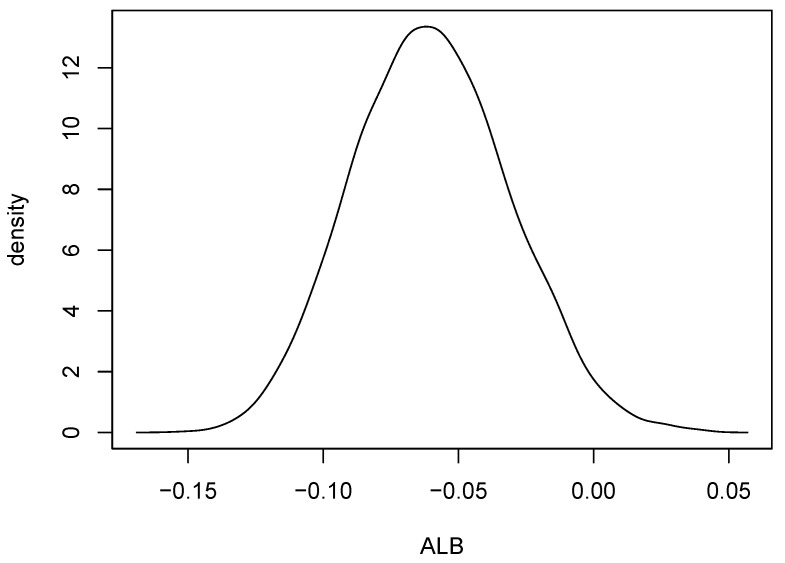
A kernel density estimate computed using 10,000 values of ALB from permuted data sets. The value of ALB for the original data set was 0.013.

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
