# Peer review of "A Bayesian Motivated Two-Sample Test Based on Kernel Density Estimates"

_entropy, 2022, doi:10.3390/e24081071_

Round 1
Reviewer 1 Report
The paper presents a new nonparametric test for the null hypothesis that two acupuncture points are equal. An attractive property of this test is that the envelope distribution defines its critical value. Such an idea is reasonable.
Positives
- The paper compares the classical two-sample problem for two distributions and considers a special case of this approach. The data in part A include all available observations except one, which is a reasonable idea.
Negatives
- The summary of the content does not feel detailed.
- The paper can be improved in the organization to follow more easily.
The following are a few issues that need further clarification and improvement.
1. The paper also needs to improve the speed of computation of the test statistic. Speed testing statistics and their reciprocal distributions, especially for large data sets.
2. The test statistic is calculated, and then select variables whose statistic exceeds a certain threshold. How do you determine the above threshold?
3. Is the random data set in the simulation credible, and is there any reference to other literature.
Reviewer 2 Report
This a very interesting papaer that is written in a clear and engaging manner. A new nonparametric test of the equality of two probability density functions is introduced. In a small simulation study, the size and power of the test are briefly illustrated and compared with competing methods. The method is extended to bivariate densities, and an application to real data is described, with graphical illustration.
At p. 19, para. 2 it would be useful to state what the 'first two variables' are.
The 'action' in figures 9-11 is rather concentrated in the bootom right hand corners. It would help to redefine the axis limits to provide more detail. Also, the addition of axis labels would be nice.
